# Inverted shear-strain magnetoelastic coupling at the Fe/BaTiO₃ interface from polarised x-ray imaging

Francesco Maccherozzi [1] ✉, Massimo Ghidini [1,2,3], Mary Vickers[3], Xavier Moya [3], Stuart A. Cavill [4], Hebatalla Elnaggar [5,6], Anne D. Lamirand [7], Neil D. Mathur [3] & Sarnjeet S. Dhesi [1] ✉

The elastic degree of freedom is widely exploited to mediate magnetoelectric coupling between ferromagnetic films and ferroelectric substrates. For epitaxial Fe films grown on clean BaTiO₃ substrates, shear strain can determine the underlying magnetoelastic coupling. Here, we use PhotoEmission Electron Microscopy of ferroic Fe and BaTiO₃ domains, combined with micromagnetic simulations, to directly reveal an inverted interfacial magnetoelastic coupling in the low-dimensional limit. We show that the magnetocrystalline anisotropy competes with the epitaxial shear strain to align the local magnetization of ultrathin Fe films close to the local polarization direction of the ferroelectric BaTiO₃ in-plane domains. Poling the BaTiO₃ substrate creates *c*-domains with no shear strain contribution with the local magnetization rotated by ~45°. Tuning shear strain magnetoelastic contributions suggests new routes for designing magnetoelectric devices.

Electrically manipulating ferromagnetic (FM) domains remains key to developing a new class of energy efficient, high-density data storage devices. Multiferroic materials are good candidates for this since they combine coupled charge and magnetic long-range ordering[1,2]. However, single-phase multiferroic materials are rare[3], due to the competing requirements of the ferroic ordering, and typically have ordering temperatures well below room temperature. To overcome such limitations, composite FE - FM heterostructures, are nowadays engineered[4,5] to exhibit multiferroic properties arising from strain coupling through ferroelastic (FEL) domains[6,7], or via exchange bias[8–10] well above room temperature. For example, the macroscopic magnetic properties of FM oxide films grown on FE substrates can be manipulated by either thermal cycling through a structural phase transition[11–13] or by applied electric fields[14]. More recently, ultrathin polycrystalline Fe films on BaTiO₃ (BTO) have been shown to exhibit strain driven superferromagnetism[15].

Ultrathin FM films grown on single-crystals can exhibit large magnetoelastic (MEL) coupling[16,17] leading to giant strain-mediated magnetoelectric effects. In this respect, the BTO(001) surface is of particular interest because of a small lattice mismatch (1.6%) with *bcc*-Fe, ensuring epitaxial growth of fully strained FM films, and because BTO is FE at room temperature. To date, Fe films with some degree of oxidation have been grown on substrate clamped BTO films allowing electrical modification of the FM overlayer[18]. However, growth of ultrathin Fe overlayers on BTO single-crystal surfaces has remained elusive due to interfacial oxidation[14]. We have developed a UHV in-situ surface cleaning and film deposition protocol that avoids the appearance of any detectable spectroscopic signature of interfacial film oxidation.

## Results and Discussion

In recent years, PhotoEmission Electron Microscopy (PEEM) has shown how magnetic domains and domain boundaries in a range of FM thin

---

[1]Diamond Light Source, Harwell Science and Innovation Campus, Didcot, UK. [2]Department of Mathematics, Physics and Computer Science, University of Parma, Parma, Italy. [3]Department of Materials Science, University of Cambridge, Cambridge, UK. [4]School of Physics, Engineering and Technology, University of York, York, UK. [5]Debye Institute for Nanomaterials Science, Utrecht University, Utrecht, Netherlands. [6]Institute of Mineralogy, Physics of Materials and Cosmochemistry, CNRS, Sorbonne University, Paris, France. [7]Ecole Centrale de Lyon, INSA Lyon, CNRS, Universite Claude Bernard Lyon 1, CPE Lyon, INL, UMR5270, Ecully, France. ✉e-mail: francesco.maccherozzi@diamond.ac.uk; dhesi@diamond.ac.uk

films, grown on BTO, evolve with an applied electric field[6,7,19–21]. In addition, PEEM together with Low-Energy Electron Microscopy (LEEM), enables detailed monitoring of thin film growth on BTO surfaces, allowing excellent control of interface composition and morphology. Here, we have successfully deposited epitaxial metallic Fe films on atomically clean (001) FE single-crystal surfaces of BTO. PEEM, combined with X-ray Magnetic Circular Dichroism (XMCD) allowed access to the Fe FM domain structure and PEEM combined with X-ray Linear Dichroism (XLD) allowed simultaneous access to the underlying BTO FEL domain structure. By interpreting the XMCD-PEEM images with a macrospin model and micromagnetic simulations, the value of shear-strain ($B_2$) magnetoelastic constant has been determined.

The BTO surface exhibited a sharp (1 × 1) Low-Energy Electron Diffraction (LEED) pattern (figure SM1) after repeated cycles of oxygen-assisted annealing. Four samples were grown for the study. Sample A had a 2.2 nm Fe film grown on the BTO, whereas Sample B, C and D had 1.5 nm films with 2 nm Al caps to allow switching of the FE domains. Mirror Electron Microscopy (MEM) of the cleaned BTO surface revealed the presence of FEL and FE domains (figure SM3) demonstrating that the surface FE properties persisted after the surface preparation. Fe films were subsequently deposited in situ with X-ray Absorption Spectroscopy (XAS) indicating the growth of purely metallic Fe (figure SM5)[22].

Figure 1 a shows an XLD-PEEM image of the Fe covered BTO surface recorded at the Ti $L_3$-edge (figure SM4). The dark/light areas correspond to $a_1/a_2$ FEL domains with the domain boundaries running parallel to the $[1\bar{1}0]_{pc}$ direction[21]. The local $c$-axis of the FEL domains is indicated by the yellow arrows (figure SM6). Figure 1b shows an XMCD-PEEM magnetization vector map (see Supplementary Material V and figure SM7 and figure SM8), recorded at the Fe $L_3$-edge, from the same area demonstrating that the Fe film FM domain structure reproduces the underlying BTO FEL domain structure. The angular distribution of the Fe magnetisation across the vector map is shown in Fig. 2a and indicates a predominant direction of 54° ± 3° for the blue areas, *i.e.* for the BTO $a_1$ domains, and 129° ± 4° for the magenta areas, *i.e.* for the BTO $a_2$ domains. The schematic in Fig. 2a shows the orientation of the BTO unit cell for each of the FEL domains so that it is straightforward to see that the local Fe spin-axis unexpectedly aligns close to the local BTO $c$-axis, confirming the presence of a strong local uniaxial magnetic

anisotropy. Such an alignment is in stark contrast to the case for thicker Fe films grown on BTO[7] which exhibit a perpendicular in-plane magnetic alignment with the local BTO $c$-axis.

The MEL coupling of the Fe film, arising from the reduced dimensionality, can be developed in terms of a macrospin model and using micromagnetic simulations. Within a macrospin (single domain) approximation, the total MAE of a cubic system can be expressed in the form of a power series[23] as the sum of a uniaxial MEL energy ($F_{me}$) and a cubic magnetocrystalline (MC) energy ($F_{mc}$) as

$$F_{tot} = F_{me} + F_{mc} = B_1 \sum_{i=j} \epsilon_{ij}\alpha_i^2 + 2B_2 \sum_{i\neq j} \epsilon_{ij}\alpha_i\alpha_j + K_1 \sum_{i\neq j} \alpha_i^2\alpha_j^2 \quad (1)$$

where $\alpha_i$ and $\alpha_j$ are direction cosines of the in-plane magnetisation, **M**, $\epsilon_{ij}$ are strain tensor components, $B_1$ and $B_2$ are the first order MEL constants, and $K_1$ is the first order MC constant. We define the angle between **M** and $[100]_{Fe}$ as $\gamma$ (hence $\alpha_1 = cos(\gamma)$ and $\alpha_2 = sin(\gamma)$). The unpoled BTO substrate exhibited *in-plane* FEL $a_1$ and $a_2$ domains. Using bulk elastic stiffness values[16], the Fe strain tensor $\epsilon$ has components $\epsilon_{11} = \epsilon_{22} = -0.0112$, $\epsilon_{33} = 0.013$ and $\epsilon_{12} = \epsilon_{21}$ which is 0.0055 for an $a_1$ domain and -0.0055 for an $a_2$ domain. Given that the magnetization is in-plane ($\alpha_3 = 0$) and that $B_1$ is independent of $\gamma$, $F_{tot}$ can be expressed in a compact form as

$$F_{tot} = K_1(r\,\alpha_1\alpha_2 + \alpha_1^2\alpha_2^2), \qquad r = \frac{2B_2\epsilon_{12}}{K_1} \quad (2)$$

The stable minima, with respect to $\gamma$ (figure SM9) arise from the competition between the shear-strain uniaxial and MC biaxial anisotropies and are plotted as a function of $r$ in Fig. 2b. The magnetic anisotropy is uniaxial for $r \leq -1$ ($r \geq 1$), with the easy-axis aligned along $[110]_{Fe}([1\bar{1}0]_{Fe})$ yielding $a_2$ ($a_1$) FEL domains, and represents the case for strained Fe/BTO using bulk *bcc*-Fe values of $K_1$ and $B_2$ ($K_1 = 0.042$ $MJ/m^3$ and $B_2 = 7.83$ $MJ/m^3$ (*i. e.* $|r_{bulk}| = 2.1$). For $|r| < 1$, the competition between the shear-strain MEL and MC anisotropies rotates the easy-axis of magnetisation from the in-plane $<100>_{Fe}$ directions to close to the $<110>_{Fe}$ directions as $r$ changes from 0 to ± 1.

The values of $\gamma$ for the $a_1(a_2)$ FEL domains, determined from the polar plot, are marked as the solid squares in Fig. 2b and show that the

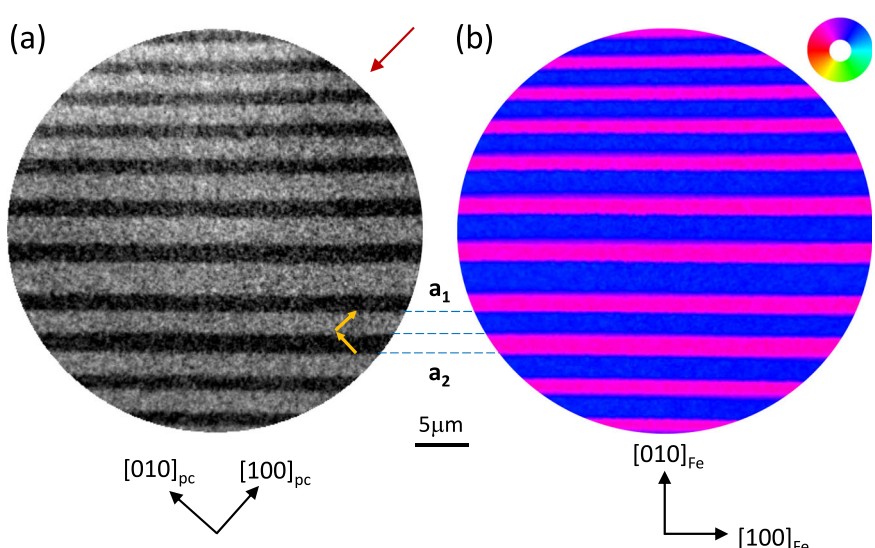

$a_1$

$a_2$

5μm

$[010]_{pc}$   $[100]_{pc}$

$[010]_{Fe}$

$[100]_{Fe}$

**Fig. 1 | PEEM images of the FEL/FM striped domains. a** XLD-PEEM image of the Fe/BTO surface recorded at the Ti $L_3$-edge showing the FEL domain structure. The $c$-axes of the $a_1$ and $a_2$ FEL domains are shown as the yellow arrows and were determined using the angle-dependence of the XLD contrast. The red arrow indicates the in-plane projection of the incoming x-rays. **b** XMCD-PEEM vector magnetisation map recorded at the Fe $L_3$-edge showing the FM magnetic domain structure of the Fe overlayer. The magnetization direction is represented by the colour wheel. The crystallographic directions are shown in black. The broken blue lines are guides to the eye linking the BTO $a_1$ and $a_2$ FEL domains with the corresponding Fe FM domains. (Data from Sample A).

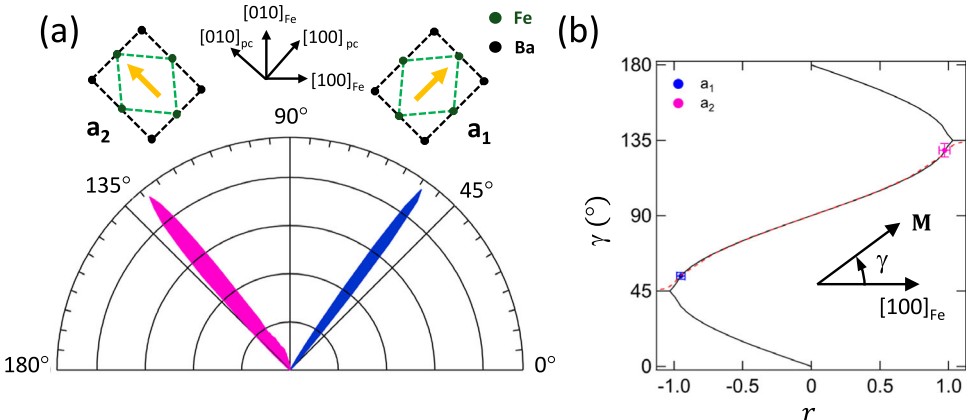

**Fig. 2 | Angular distribution of the magnetization direction. a** Polar plot of the angular distribution of the local spin-axes extracted from the vector magnetisation map shown in Fig. 1b along with a representation of the corresponding Fe/BTO unit cells for the $a_1$ and $a_2$ FEL domains. The yellow arrows indicate the local $c$-axis direction. The crystallographic axes are shown in black along with the BTO pseudocubic (pc) and Fe unit cell orientations. **b** Stable values of the magnetisation angle, $\gamma$, determined using a macrospin model (solid black line) and micromagnetic simulations (broken red line) as a function of $r$. The experimental values of $\gamma$, determined from the polar plot for the $a_1/a_2$ FEL domains, are marked as the solid blue/magenta squares. Error bars represent the FWHM of the polar plot peak widths. (Data from Sample A).

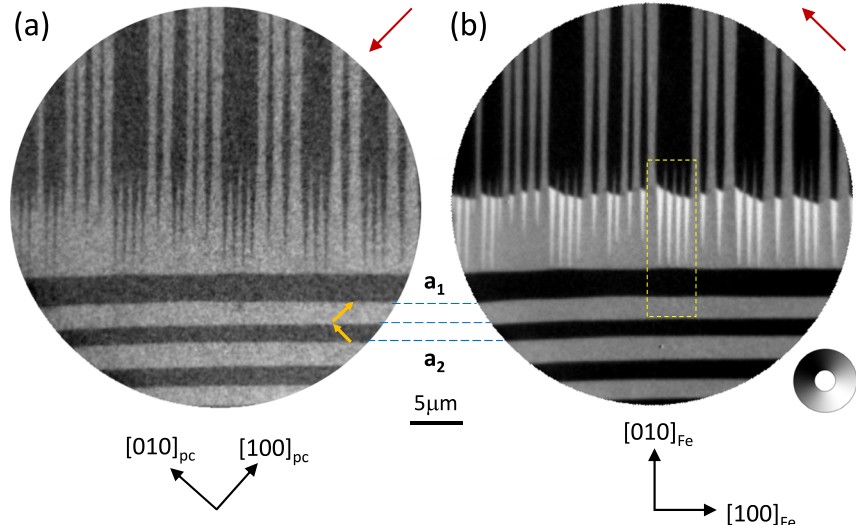

**Fig. 3 | PEEM images of the FEL/FM needle domains. a** XLD-PEEM image of the Fe/BTO surface recorded at the Ti $L_3$-edge showing the FEL domain structure of an area where the DWs run along the $[110]_{pc}$ direction (top half) and meet DWs running along the $[1\bar{1}0]_{pc}$ direction (bottom half) forming a well-defined tapered FEL domain structure. The yellow arrows indicate the local BTO $c$-axis. **b** XMCD-PEEM image of the same area as (a) recorded at the Fe $L_3$-edge showing the corresponding FM domains. The magnetisation direction is represented by the greyscale wheel. The crystallographic axes are shown in black and the in-plane projection of the incoming x-ray beam is shown by the red arrows. The broken yellow lines indicate the area compared to the micromagnetic simulations in Fig. 4a. (Data from Sample A).

measured local magnetisation direction is $8 \pm 2°$ from the $<110>_{Fe}$ ($<\bar{1}10>_{Fe}$) directions, implying a negative value of $B_2$ and representing a significantly different MEL coupling in ultrathin Fe films grown on BTO compared to thicker films.

In order to take into account the dipolar interactions between the FM domains, micromagnetic simulations were performed. The dashed line in Fig. 2b shows the resulting stable minima using the bulk $bcc$-Fe value for $K_1$ and assuming an initial Fe magnetic configuration with $\mathbf{M} \parallel [\bar{1}10]_{pc}$. Figure 2b shows good agreement between the dependence of $r$, with $\gamma$, extracted from the micromagnetic simulations and calculated using the macrospin model, with the exception of the transition regions ($|r| \approx 1$) where there is a small difference between the two curves. By averaging over the FEL domains, $r$ is determined to be $-/+ \, 0.97 \pm 0.02$ for the $a_1/a_2$ domains. However, quantitatively extracting the effects of shear strain from $r$ requires an accurate determination of $K_1$ which depends sensitively on film thickness[24]. To address this uncertainty, the changes in the Fe domain pattern with $r$ can be explored using micromagnetic simulations, but then requires sharper domain features to uniquely evaluate both $B_2$ and $K_1$ from the domain pattern transfer.

Figure 3a shows an XLD-PEEM image with a pattern of dark/light areas arising from the $a_1/a_2$ FEL BTO domains, with long domain walls parallel to either $[1\bar{1}0]_{pc}$ or to $[110]_{pc}$ as observed for the FEL domains shown in Fig. 1a. The tapering of the FEL domain walls, also observed using MEM (figure SM3), is driven by strain minimisation[25] at the boundaries between orthogonal FEL domains. The analysis of the contrast in the XLD-PEEM image confirms an *in-plane* orientation of the local FEL domain $c$-axis (shown as the yellow arrows). Figure 3b shows an XMCD-PEEM image of the same area as Fig. 3a with the contrast showing the projection of the local magnetisation onto the light propagation direction. Inside the tapered $a_2$ domains, black and white regions with sharp boundaries arise from magnetic domains with

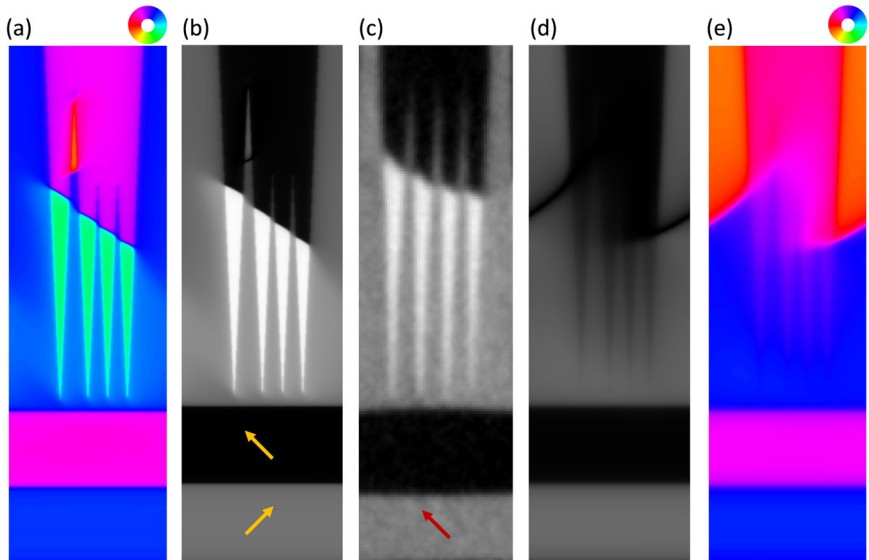

**Fig. 4 | Micromagnetic simulations of the PEEM images. a** Micromagnetic simulation of the Fe magnetic domains with $B_2 = -3\,MJ/m^3$ ($|r| = 0.97$) corresponding to the area shown by the broken yellow lines in Fig. 3b. The magnetisation direction is represented by the colour wheel. **b** Simulated XMCD-PEEM image for the domain structure shown in (**a**). The yellow arrows indicate the the local BTO $c$-axis. **c** Experimental XMCD-PEEM image of the tapered FM domains highlighted by the yellow broken-line box in Fig. 3b. The red arrow indicates the in-plane projection of the incoming x-rays. **d** Simulated XMCD-PEEM image and (**e**) corresponding magnetic domain structure for $B_2 = -0.2\,MJ/m^3$ ($|r| = 0.97$) showing the degraded transfer of the domains structure. The short side of the images is $4\,\mu m$.

head-to-head domain walls. Inside the $a_1$ regions the XMCD contrast is grey, consistent with a magnetization direction perpendicular to the x-ray beam direction. The uniformity of the XMCD contrast suggests a strong uniaxial anisotropy within the FEL domains.

To quantitatively understand the interfacial MEL coupling, micromagnetic simulations were performed with $|r| = 0.97$. By setting a uniform initial state with $\mathbf{M} \parallel [\bar{1}\bar{1}0]_{pc}$, the simulations converged to a domain structure which included the head-to-head DWs observed on the needle tip structures within the $a_2$ FEL domains (see Fig. 4a). Figure 4b shows the corresponding simulated XMCD-PEEM image along with the corresponding experimental XMCD-PEEM image in Fig. 4c. The simulations also predict the formation of 180° DWs in the tapered $a_1$ FEL areas which are not observed in the experimental XMCD-PEEM images due to the experimental geometry and the resolution limit of the field of view used. Reducing $B_2$ to $-0.2\,MJ/m^3$ (whilst keeping $r$ constant) results in a significant blurring of the FM domain structure (see Fig. 4d, e). By comparing the experimental and simulated XMCD-PEEM images for a range of $B_2$ values, $B_2$ is estimated to be $-3 \pm 1MJ/m^3$ with $M_{sat} = 1.76\,MA/m$ (figure SM10) and $0.022 < K_1 < 0.042\,MJ/m^3$ (figure SM11) and represents the first quantitative determination of the interfacial shear strain MEL coupling in a FM thin film.

Thin film MEL constants have also been determined using a Néel model with the addition of surface contributions to represent the MAE of a thin film[16,17,26]. Another possibility is to include higher order terms to the MAE[24,27,28] using a strain-dependent effective MEL constant, $B_{2,eff}$ (figure SM12), which can reverse sign for thin films under strain. Although such a result might imply a negative value of $B_2$ for the Fe/BTO structure here, the similarity between $B_{2,eff}$ and $B_2$ is not direct and each coefficient describes distinct physical quantities. $B_{2,eff}$ is proportional to the anisotropic *stress* that a FM film exerts on a substrate as $\mathbf{M}$ is rotated in-plane, and represents the strain derivative of the MAE. Conversely the local direction of $\mathbf{M}$, with the in-plane strain state determined by the epitaxial growth, is governed *directly* by the angular dependence of the MAE. We verified that adding the experimentally determined higher order terms to $F_{me}$ did not have an effect on the easy-axis direction of the Fe thin film (figure SM12). The emergence of a negative value of $B_2$ in Fe/BTO heterostructures then highlights the importance of epitaxial shear-strain

in thin films which depends sensitively on factors such as morphology and interface quality[16].

In order to further investigate shear strain effects on the ME coupling at the Fe/BTO interface, samples capped with 2nm of Al were prepared so that the BTO substrate could be poled by applying an electric field of 200 kV/m between the top and bottom sample. Once poled, the capped samples exhibit $c$ domains for which the shear strain should be zero with only the MC anisotropy remaining. Figure 5a shows a composite XMCD-PEEM and XLD-PEEM image before poling (figures SM13 and SM15) with the local $\mathbf{M}$ at $58° \pm 2°$ from the local BTO $c$-axis for the (blue) $a_2$ FEL domains and $74° \pm 1°$ from the local BTO $c$-axis for the (magenta) $a_1$ FEL domains. This yields $B_2 = 2.5 \pm 0.3\,MJ/m^3$, which is significantly smaller than than the $bcc$-Fe bulk value with the change of sign, with respect to the uncapped Fe film, likely due to a large reduction of the shear strain due to the Al capping layer. Figure 5b shows a composite image with a $c$ domain, created after poling ($40\,MV/m$) the sample. The XMCD-PEEM vector map shows that the introduction of the FEL $c$ domain precipitates a concomitant reorientation of the local Fe magnetisation (figures SM14 and SM16). Figure 5c shows the angular distribution of $\mathbf{M}$ in the XMCD-PEEM vector map, before and after poling, indicating that the ME coupling on the $a_2$ domain remains largely unaffected. Conversely, the Fe magnetisation is almost parallel to the $[010]_{Fe}$ direction in the vicinity of the new FEL $c$ domain. Poling the BTO substrate therefore demonstrates that, in the absence of a local shear strain, only the MC anisotropy remains relevant which aligns $\mathbf{M}$ close to the expected bulk $bcc$-Fe easy axes (table SM1). Ferromagnetic resonance (FMR) performed on sample D yielded a value of $B_2$, lower than that for bulk $bcc$-Fe, but compatible with the PEEM results (figure SM17).

Shear strain effects at a FM-FE interface have been quantified using FEL and FM domain structures in Fe ultrathin films grown on BTO using polarized x-ray based PEEM imaging. The Fe film exhibits strain-mediated FEL-FM domain transfer down to the smallest resolvable features ($200\,nm$). The sharp alignment of the local Fe magnetisation direction with the local FEL $c$-axis, combined with micromagnetic simulations, enabled the detection of a change in magnitude and sign of the MEL coupling constant, $B_2$, with respect to thicker Fe films or films with an Al capping. Our results reveal new routes to

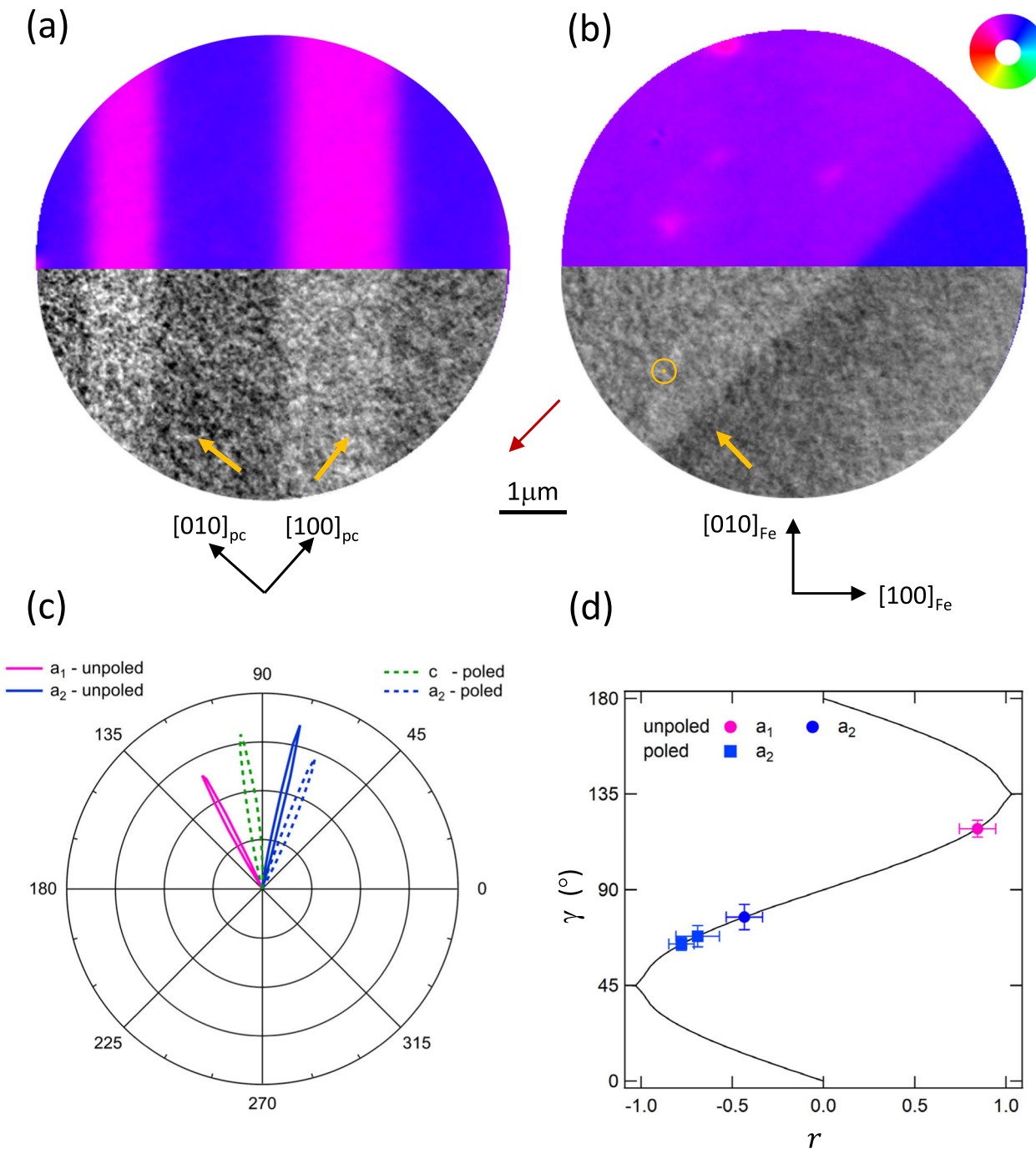

**Fig. 5 | Poling the BTO substrate.** Composite images of the (**a**) unpoled and (**b**) poled Al/Fe/BTO surface comprising an XMCD-PEEM vector map of the Fe magnetization (top half) and XLD-PEEM of the FEL domains (bottom half). The red arrow represents the in-plane projection of the x-ray beam. The FEL domain *c*-axis orientation is indicated in yellow for each domain and the magnetization direction is represented by the colour wheel. The crystallographic axes are shown in black. **c** Polar plot of the magnetization angle, $\gamma$, distribution for the unpoled (solid lobes) and poled (broken lines) states. **d** $\gamma$ as a function of $r$ for stable minima of the magnetic free energy (solid black line). The experimental values of $\gamma$, determined from the polar plot, are marked as circles/squares for the unpoled/poled states and as magenta/blue symbols for the $a_1/a_2$ FEL domains. Error bars represents the FWHM of the magnetization distribution peaks. (Data from Sample C).

understanding MEL coupling in composite heterostructures which combine inverse magnetostriction in thin FM films with shear strain effects from epitaxial growth.

## Methods
### Samples
At room temperature, BTO ($a_{BTO} = 3.992$ Å and $c_{BTO} = 4.036$ Å) exhibits a FE tetrahedral phase with the local *c*-axis of the FEL domains oriented in- and out-of-plane. A pseudocubic (*pc*) notation has been used

throughout this work when referring to the BTO crystal structure. *bcc*-Fe ($a_{Fe} = 2.87$ Å) grows with an epitaxial relation $[110]_{Fe}||[100]_{pc}$ on BTO (figure SM1).

The BTO crystals were supplied by Crystec Gmbh and SurfaceNet Gmbh with surface mechanical polishing and $HNO_3$ chemical etching performed by Crystec Gmbh (rms roughness <0.5nm). Unpoled BTO(001) substrates were cleaned by repeated oxygen-assisted annealing cycles ($T = 630°$, $P(O_2) = 2 \times 10^{-6}$ mbar). The procedure limited heat-induced reduction of the surface while desorbing charge-

screening contaminants which can mask the surface electric dipoles. The structure of the BTO surface was monitored using LEED while the BTO FEL and FE domain structures were monitored using LEEM in MEM mode (figure SM3). *bcc*-Fe films were deposited by *e*-beam evaporation (0.1 Å/min) with a thickness, determined using a quartz microbalance. The film thickness was confirmed using ex situ X-ray Reflectivity for sample B and C and X-ray Diffraction for sample A. Sample thicknesses were 2.2 nm for sample A and 1.5 nm for samples B, C and D. During Fe deposition, the BTO substrate was held at 300 °C, i.e., above the paraelectric to ferroelectric phase transition, so that the BTO FEL and Fe FM domain structure formed as the substrate cooled to room temperature. Samples B, C and D were each capped with 2nm of Al to allow in situ poling. Capping with low atomic number elements, such as Al which have lower X-rays absorption coefficients and generate fewer secondary electrons, results in an improved XAS signal to noise level.

### PEEM imaging
XMCD-PEEM and XLD-PEEM imaging were performed on beamline I06 of Diamond Light Source using an Elmitec SPELEEM III equipped with LEEM. PEEM images were obtained at room temperature in zero applied magnetic field using secondary electrons arising from the absorption of polarised X-rays incident at 16° to the sample surface. XMCD-PEEM images of the Fe FM domains were acquired using circularly polarised x-rays at photon energies, $E_1 = 706.6$ eV ($E_2 = 702.8$), corresponding to on(off) the Fe $L_3$-edge resonance. XLD-PEEM images of the BTO FEL domains were acquired using linearly polarised x-rays at photon energies, $E_1 = 456.65$ eV ($E_2 = 454$ eV), corresponding to on(off) the Ti $L_3$-edge resonance. The Ti on-resonance energy corresponds to the first peak of the $L_3$-edge XLD spectrum (figure SM4). The XMCD/XLD contrast was calculated for each image pixel as the difference in intensities divided by the sum of intensities, for two different polarisation states of the x-rays. Sets of 200/400 PEEM images (1s/image) were averaged to obtain XMCD/XLD contrast of the FM/FEL domains. FM vector maps were constructed by combining two XMCD-PEEM images taken using orthogonal sample orientations (figure SM8), after correcting for drift and distortion via an affine transformation using non-dichroic features on the XAS images.

### Micromagnetic simulations
Micromagnetic calculations were performed using Mumax3[29] to simulate the FM domain structures on the $a_1$ and $a_2$ FEL domains. The cell - grid sizes were $15.6 \times 15.6 \times 1.5$ nm - $256 \times 1280 \times 1$ pixel ($1.95 \times 1.95 \times 1.5$ nm - $1536 \times 1536 \times 1$ pixel) for the needle (striped) domain patterns, respectively. Periodic boundaries conditions were applied to the horizontal direction for the needle pattern and to both in-plane directions for the striped pattern using 4 virtual repetitions of the grid. The spatially modulated in-plane strain arising from the FEL domains was simulated using a local uniaxial energy term and set parallel to the $[100]_{pc}$ ($[010]_{pc}$) direction for the $a_1(a_2)$ domain areas. The initial magnetic configuration was set with $\mathbf{M}||[\bar{1}10]_{pc}$ with an exchange energy, $A = 2 \times 10^{-11} J/m$. The in-plane grid size (15.6 nm) was larger than the calculated exchange length, $l_{ex} = 2.3$ nm[30]. We verified that reducing the cell size down to 2 nm did not change the results significantly. Both random and collinear initial magnetic configurations were tested with the latter presented here. The initial configuration was left to dynamically evolve for 4 ns and then allowed to relax to its lowest energy configuration. The effects of changing $B_2$, whilst keeping $r$ constant, were explored as well as the effects of changing $M_{sat}$ (figure SM10). The XMCD-PEEM images were calculated from the simulated magnetization maps.

### Ferromagnetic resonance
In order to measure the in-plane anisotropic ferromagnetic resonance (FMR) we utilized a broadband modulation setup in combination with a 2D vector magnet[31] that measured the derivative of the FMR absorption, $\frac{dP}{dH}$, via lockin detection.

### X-ray diffraction
X-ray diffraction was performed using a four-circle high-resolution Panalytical Empyrean diffractometer ($Cu - K_{\alpha 1}$, 1.5406 Å). Data were obtained by placing either a long parallel collimator (LPC) or a beam tunnel in front of the detector (figure SM2). At the cost of lower intensity, the LPC allows more accuracy in determining peak positions since the acquired data are insensitive to any possible error in sample height.

## Data availability
The datasets supporting the findings of this study are openly available via figshare at: https://doi.org/10.6084/m9.figshare.29041892.v1 (2025).

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

## Acknowledgements

We thank Diamond Light Source for the provision of beamtime under proposal numbers SI-12893 and NT-5888. We thank Adriano Verna for performing the X-Ray Reflectivity (XRR) experiments. MG gratefully acknowledges support from the University of Parma through the action Bando di Ateneo 2021 per la ricerca co-funded by MUR-Italian Ministry of Universities and Research - D.M. 737/2021 - PNR - PNRR - NextGenerationEU. X. M. is grateful for support from the Royal Society.

## Author contributions

F.M. and S.S.D. conceived the project and performed the PEEM and LEEM experiments with help from A.D.L., M.G., X.M. and N.D.M.; F.M. and S.S.D. performed the micromagnetic simulations. H.E. performed the multiplet calculations. S.A.C. performed the FMR measurements, analysis and simulations. M.V. performed the XRD measurements. S.S.D. and F.M. wrote the manuscript with contributions from all co-authors.

## Competing interests

The authors declare no competing interests.
