## [Transparent Peer Review file · Nature Communications]

Inverted shear-strain magnetoelastic coupling at the Fe/BaTiO₃ interface from polarised x-ray imaging

Corresponding Author: Dr Sarnjeet Dhesi

Version 0:

Reviewer comments:

Reviewer #1

(Remarks to the Author)

Francesco Maccherozzi et al. utilized synchrotron radiation based techniques to study the Fe/BTO bilayer with multiferroic coupling. As stated, this work found that the higher order terms from magnetoelastic coupling cannot explain the observed magnetic domain structure, and the shear strain was introduced to be important. The work indeed provide some evidence to support their argument, however, some key information is still missed. The detailed comments is as below:

1. The FE domain of BTO is believed to coupling with FM domain of Fe film with strain, and the equations was employed to predict the phenomena. Based on the discussion, the Fe is supported to epitaxially grown on BTO, and the direct experimental evidence is required. Polycrystal might be present, which would relax strain around the grain boundary.
2. The vector magnetization map is used to statistically study the distribution of magnetization. More details on how to overlap two images in Fig. SM6a-6b, such as how to tune the contrast and intensity of the figure. what is the resolution in direction angle?
3. In the whole manuscript, the FE polarization is along either a₁, a₂ or c domains. Similarly as Fe film, is this technique able to grab the gradually change of FE polarization.
4. Fig. 4 study the FM domain structure with 180 domain wall around needle tip structures, and Fig. SM2 claims the FE domain structure with 180 domain wall with MEM image in similar area. Does this coincidence appears randomly or imply any coupling between them?
5. Fig. 1 and Fig.5 Both use the colour wheel to represent the direction of magnetization of a₁ and a₂ area. However, This is a discrepancy about the magnetization direction between them. For example, for a₁ area the magnetization is right-up in Figure 1, while it is left-up in Figure 5. In addition, the axis label is also different between them, making it confused to compare.
6. After poling, why a₂ domain is still present while the a₁ domain disappears?
7. For Fe film on c domain, is the interfacial charge effect important to determine the magnetic anisotropy in Fe film? Forming 90/180 FM multidomain?
8. As stated in the manuscript, the direction of magnetization of Fe film is different from previously report work, which is attributed to the different in Fe thickness. More direct evidence should be provided.
9. what is the conductivity of 1.5 nm Fe film? If the film is continuous with good conductivity, the Al deposition might be not necessary.

Reviewer #2

(Remarks to the Author)

The authors claim to observe the anomalous interface magnetoelastic coupling in ultrathin Fe film on BaTiO₃ (BTO) by PhotoEmission Electron Microscopy (PEEM) imaging. Specifically, they found the alignment of the local Fe magnetization direction with the local BTO c-axis in ultra-thin Fe film grown on BTO, which is in contrast to the case for thicker Fe films. The authors have concluded that the result originates from the change in the sign of the magnetoelastic constant (B₂). In addition, they provide a quantitative determination of the values of B₂ and the magnetocrystalline anisotropy constant (K₁). The obtained results are interesting and noteworthy, and I believe the present study is suitable for publication in Nature Comms. However, I have a few questions that I hope the authors can address in a revised version of the manuscript.

1. The abstract claims that the competition between normal and shear strains aligns the local magnetization with the local polarization. However, in the main manuscript, the discussion is presented from the viewpoint of the competition between shear-strain-induced uniaxial anisotropy and magnetocrystalline (MC) biaxial anisotropy. This discrepancy in the description is confusing.

2. Why is the 2-nm-thick Al layer capped to enable the switching of the FE domains?

3. I hope the authors comment on the thickness dependence of the sign of B₂. I wonder up to what Fe thickness the negative B₂ persists. Does the negative sign of B₂ remain even above an Fe thickness of 1.5 nm?

4. The authors describe: "Here, we have successfully deposited 1.5 nm of metallic Fe on atomically clean (001) FE single-crystal surfaces of BTO." What is the key factor in achieving a high-quality Fe film? Comments on this point would be helpful for the readers.

Reviewer #3

(Remarks to the Author)

This work reports an unexpected competition between normal and shear strain in manipulating magnetoelastic coupling. The authors primarily rely on XLD-PEEM and simulations to support their conclusions. However, there are several points that remain unclear in the manuscript, which diminish its potential for publication in Nature Communications.

1. The manuscript lacks details regarding the deposition of the Fe films. How were the Fe films deposited, and how was their thickness controlled? What method was used to maintain their metallic properties, particularly to prevent oxidation at a thickness of 1.5 nm? These details are crucial to the work but are largely omitted from the manuscript.

2. There is no explanation provided for why a thickness of 1.5 nm was chosen. On page 5, the authors mention that the behavior within a 1.5 nm Fe film differs from that of thicker films. However, they do not present any data on how the thicker Fe films behave. Furthermore, it is unclear how "thick" and "thin" are defined in this context—what is the boundary between them? It is also suggested that the authors investigate the effect of film thickness in relation to the proposed mechanism in the current work.

3. The experimental investigations appear to focus primarily on plane-view observations, with a lack of cross-sectional characterization of the samples. TEM could be employed to visualize stress or domain propagation, which would provide additional support for the authors' simulated results.

4. The manuscript would benefit from more effort in terms of organization, as the lack of subsection divisions makes it a bit harder to follow.

5. The authors may consider measuring some macroscopic magnetoelectric coupling parameters to support their discussion, as they specifically mention the magnetoelectric coupling effect or coefficient in the introduction with regard to potential applications.

6. One might wonder how the electrode was deposited with a thickness of 2 nm, and whether this introduces any variables related to stress. Did the authors compare this electrode to others, such as Pt? This comparison could help exclude any additional variables introduced by the electrode, especially since it has the same thickness as the Fe film.

7. I believe the authors mention the polycrystalline nature of the Fe films. However, in the conclusion paragraph (page 10), they summarize the manuscript by referring to epitaxial growth, which is quite confusing. This conclusion is difficult to follow, especially in the context of the key highlights of the current work. Additionally, how should the epitaxial growth of metals on perovskite inorganic materials be defined? Are the Fe films polycrystalline, epitaxial, or amorphous? There is a lack of convincing discussion on this point.

Version 1:

Reviewer comments:

Reviewer #2

(Remarks to the Author)

The authors have adequately addressed the comments raised in the previous review. I am satisfied with the revised manuscript and recommend it for publication in Nature Communications.

Reviewer #3

(Remarks to the Author)

The authors addressed all concerns, the reviewer suggest its publication.

Responses to Reviewer #1

We thank the referee for reviewing our work and for their recommendations to add further information. We hope that, with the changes we have made and with the additional data that we have presented, all the points raised by the referee have been sufficiently addressed. We address each point in detail below.

1. The FE domain of BTO is believed to coupling with FM domain of Fe film with strain, and the equations was employed to predict the phenomena. Based on the discussion, the Fe is supported to epitaxially grown on BTO, and the direct experimental evidence is required. Polycrystal might be present, which would relax strain around the grain boundary.

The single crystalline state of the Fe film is an important point and we thank the referee for raising this issue. We note that the crystallinity of the film surface was confirmed by Low Energy Electron Diffraction (LEED) as shown in section 1 of the Supplementary Material (SM). However, following the comments of the reviewer, we have added X-ray Diffraction (XRD) data along with a discussion of the results to section 1 of the SM. The XRD results, in combination with the LEED results, confirm the epitaxial growth of the Fe film under *in-plane* compressive strain. Furthermore, we reanalysed our XRD data to extract the film thickness and we now have an updated value for the film thickness (2.2nm cf. 1.5nm previously) which we have now used throughout the manuscript.

2. The vector magnetization map is used to statistically study the distribution of magnetization. More details on how to overlap two images in Fig. SM6a-6b, such as how to tune the contrast and intensity of the figure. what is the resolution in direction angle?

We have extended section V of the SM to clarify how we constructed the magnetisation vector maps by overlaying two XMCD images taken with the *in-plane* projection of the incident x-rays rotated by 90°. The standard deviation of the magnetisation angle is +/- 2° for a single pixel and arises from the noise in the XMCD intensities. However, the error in the magnetisation angle, shown for the different magnetic domains in *e.g.* Figure 2, was determined using the FWHM of the angular distribution (*i.e.* the width of the lobes in the polar plot shown in Fig. 2) across ~10⁴ pixels.

3. In the whole manuscript, the FE polarization is along either a1, a2 or c domains. Similarly as Fe film, is this technique able to grab the gradually change of FE polarization.

In XMCD it is possible to construct a vector map of the gradually changing direction of the in-plane magnetization, \mathbf{M} , because the XMCD contrast is proportional to component of \mathbf{M} along the x-ray beam propagation direction. The same approach cannot be applied to acquire a vectorial map of the FE *polarisation* for the following reasons.

X-ray Linear Dichroism (XLD) at the Ti L_{2,3} edges is sensitive to the ferroelastic (FEL) distortion of the crystal, which is associated with the ferroelectric (FE) polarisation, but it is not *directly* sensitive to the FE polarisation vector. Moreover, the XLD is sensitive to the FEL/FE axis orientation, but not to the direction of the FE polarisation.

We also note that XLD is sensitive to the ferroelastic crystal distortion, which is the main driver for the magnetoelastic interaction. The XLD contrast is uniform in each FEL

domain, within the noise level, so it is reasonable to assume that the ferroelastic strain axis does not deviate significantly from the local (001) c -axis direction of the BTO, with the exception of regions near the domain walls. However, any deviations at the domain walls is below the spatial resolution of PEEM.

4. Fig. 4 study the FM domain structure with 180 domain wall around needle tip structures, and Fig. SM2 claims the FE domain structure with 180 domain wall with MEM image in similar area. Does this coincidence appears randomly or imply any coupling between them?

The needle like white FM domains in Fig. 4 have some morphological similarities with the FE domain structures seen on the bare BTO(001) substrate in the MEM image in section II of the SM. However, the two domain structures have a different origin. The ferromagnetic needle domains nucleate to minimise the magnetostatic energy. A view that is supported by Fig. SM 10 which shows the dependence of the white needle domain structure on the Fe saturation magnetisation.

We also note that the FE 180° domain walls are dynamically nucleated by the MEM electron beam. These domain walls appear and move spontaneously, as the incident electron energy is varied (see SM Video 1). On the other hand, we did not observe any time evolution or displacement of Fe FM domains. In addition, the FE 180° domain walls form a zig-zag pattern (Fig. SM3) that is not present in the Fe FM domain pattern shown in Fig 4. We therefore excluded a relationship between the FE 180° domains, observed with MEM, and the needle Fe FM domains, observed using X-ray Magnetic Circular Dichroism based PEEM.

5. Fig. 1 and Fig.5 Both use the colour wheel to represent the direction of magnetization of a_1 and a_2 area. However, This is a discrepancy about the magnetization direction between them. For example, for a_1 area the magnetization is right-up in Figure 1, while it is left-up in Figure 5. In addition, the axis label is also different between them, making it confused to compare.

We thank the referee for pointing out this discrepancy. We have corrected the error and the co-ordinates are now consistent with Fig. 1.

6. After poling, why a_2 domain is still present while the a_1 domain disappears?

During poling, at the peak of the applied electric field, the BTO(001) surface becomes a single c -domain. When the electric field is released, *in-plane* a_1/a_2 domains nucleate, to reduce the elastic energy of the BTO(001) surface. The a_1/a_2 domains then rearrange in stripes of alternating a_1 - c or a_2 - c domains, oriented, respectively, along $[010]_{pc}$ or $[100]_{pc}$, in order to minimize the elastic energy. In the FOV that is presented in our manuscript, the poling process created an a_1 - c domain stripes structure. The presence of an a_2 domain alongside the a_1 - c alternating stripes is not compatible due to the large elastic energy involved. The absence of a_2 domains in the FOV presented can be explained by the $[100]_{pc}$ orientation of the domain boundaries, that allows only a_1 - c alternating domains. It is highly likely that in other regions of the sample, the a_2 - c stripe structure is present. However, our aim in poling the BTO(001) surface was to demonstrate the absence of shear-strain effects within the c -domains, and so we did not extensively explore other areas of the surface.

7. For Fe film on c domain, is the interfacial charge effect important to determine the magnetic anisotropy in Fe film? Forming 90/180 FM multidomain?

The effect of the interfacial charge is, indeed, an aspect of the ME coupling that we explored, but we have not found any evidence that it plays a role. Our model can explain all details of the FM domain formation considering only magnetoelastic coupling. Ferroelectric domains are characterized by an electric polarization and a ferroelastic strain, as discussed in our response to point 4 above. That is that XLD is sensitive to the FEL strain, but it is not sensitive to the direction of the ferroelectric (FE) polarisation and cannot image 180° FE domains. Similarly, LEEM cannot probe the 180° FE domains under the Fe film, because the surface electric field generated at the domain boundaries is screened by the metal overlayer. We don't have a way to probe the presence of 180° FE domains under the Fe film, however we have not observed the zig-zag structures associated with the 180° FE domains, as in fig. SM 3, on the Fe films on any of our samples, before or after poling. We, therefore, do not have any evidence that the FE charge plays a role in the ME coupling.

8. As stated in the manuscript, the direction of magnetization of Fe film is different from previously report work, which is attributed to the different in Fe thickness. More direct evidence should be provided.

Our work concentrated on the ultrathin film regime on bulk BTO substrates, which has not been explored before. For Fe films as thin as 0.6 nm we observed large areas with no coupling to the underlying BTO FEL domains. For Fe films thicker than 3nm, the Ti XAS signal level would have been much reduced so that imaging the underlying BTO FEL domains would have been much more difficult and so we did not explore this thickness regime.

A full thickness dependent study would require a significant amount of additional x-ray beamtime and would only reveal results up to a thickness of 4nm due the attenuation of the Ti signal with the increasing Fe thickness. Hence, and unfortunately, a thickness dependent study was outside the scope of our study, but we hope to inspire further x-ray microscopy studies now that we have established that ultrathin, metallic Fe films can be grown on BTO.

9. what is the conductivity of 1.5 nm Fe film? If the film is continuous with good conductivity, the Al deposition might be not necessary.

The Fe films are electrically conductive and allow X-ray based PEEM imaging without surface charging. However, the Fe films were grown *in situ* after the atomically clean BTO surface was prepared and characterised using LEEM. The preparation and characterisation of the BTO(001) surface under uhv conditions was key to successfully depositing thin metallic Fe films on a FE BTO surface. In order to pole the Fe/BTO structure, a different PEEM sample cartridge had to be used to apply an electric field. The Al film was deposited *in situ* to prevent oxidation of the Fe surface during exposure to air whilst the sample was transferred, in air, to a cartridge with electrical contacts to the Fe film.

Responses to Reviewer #2

We thank the referee for reviewing our work and for their positive comments on our work. We hope that, with the changes we have made and with the additional data that we have presented, all the points raised by the referee have been sufficiently addressed. We address each point in detail below.

1. The abstract claims that the competition between normal and shear strains aligns the local magnetization with the local polarization. However, in the main manuscript, the discussion is presented from the viewpoint of the competition between shear-strain-induced uniaxial anisotropy and magnetocrystalline (MC) biaxial anisotropy. This discrepancy in the description is confusing.

We thank the referee for raising this point. We now realise that the terminology in the abstract and in the conclusions is not precise enough. We have modified the text by replacing “normal” with “magnetocrystalline”, which aligns better with the conclusions of our study.

2. Why is the 2-nm-thick Al layer capped to enable the switching of the FE domains?

The samples had to be exposed to air to be mounted on a different sample cartridge that allowed the application of an electric field. The Al film was deposited *in situ* to prevent oxidation during the air exposure.

3. I hope the authors comment on the thickness dependence of the sign of B₂. I wonder up to what Fe thickness the negative B₂ persists. Does the negative sign of B₂ remain even above an Fe thickness of 1.5 nm?

Our work concentrated on the ultrathin film regime on bulk BTO substrates, which has not been explored before. For Fe films as thin as 0.6 nm we observed large areas with no coupling to the underlying BTO FEL domains. For Fe films thicker than 4 nm, the Ti XAS signal level would have been much reduced so that imaging the underlying FEL domains would have been more difficult and so we did not explore this regime.

A full thickness dependent study would require a significant amount of additional beamtime and would only reveal results up to a thickness of 4nm due the attenuation of the Ti signal with the increasing Fe thickness. Hence, and unfortunately, a thickness dependent study was outside the scope of our study, but we hope to inspire further x-ray microscopy studies now that we have established that ultrathin, metallic Fe films can be grown on BTO.

For ultrathin films, the magnitude of the magnetoelastic constants also depends on interface quality and film structure, which are not straightforward to understand and evaluate [D. Sander, Reports on Progress in Physics 62, 809 (1999)]. We grew films on BTO substrates supplied by Crystec GMBH and SurfaceNet GMBH, but only the substrates polished by Crystec allowed the growth of high-quality Fe films. We have added more information to the Methods section of the main paper to outline how the BTO substrates were prepared before insertion into the uhv system.

4. The authors describe: "Here, we have successfully deposited 1.5 nm of metallic Fe on atomically clean (001) FE single-crystal surfaces of BTO." What is the key factor in achieving a high-quality Fe film? Comments on this point would be helpful for the readers.

The main factor is the quality of the BTO surface. The mechanical polishing (performed by Crystec GmbH) was followed by etching in HNO₃, resulting in surfaces with an rms roughness of <0.5nm. In our study we were able to characterise the FE surface using LEEM-based Mirror Electron Microscopy after repeated cycles of UHV annealing in O₂. The growth temperature was set to be above the BTO ferroelectric/paraelectric phase transition and we achieved good epitaxial growth at 300°C. We have expanded the Methods section so that the reader is better able to understand the key factors in our approach to preparing the BTO surface and depositing the Fe films.

Responses to Reviewer #3

We thank the referee for reviewing our work and for their positive comments on our work. We hope that, with the changes we have made and with the additional data that we have presented, the points of concern have been sufficiently addressed. We address each point in detail below.

1. The manuscript lacks details regarding the deposition of the Fe films. How were the Fe films deposited, and how was their thickness controlled? What method was used to maintain their metallic properties, particularly to prevent oxidation at a thickness of 1.5 nm? These details are crucial to the work but are largely omitted from the manuscript.

The films were deposited by *e*-beam assisted evaporation, with the deposition rate calibrated with a quartz crystal thickness monitor. We verified the sample thickness using X-Ray Reflectivity and X-Ray Diffraction. We have now added the XRD results to section 1 of the Supplementary Material. We have also expanded the Methods section to add more details about the thin film growth.

The metallic properties were inferred from the XAS line shape presented in SM section IV. There were no multiplet features visible which precludes oxide formation. The study of the magnetic domains leading to the negative value of B_2 were performed without an Al cap as the Fe films were grown *in situ*. Once the study was complete, the films were capped with 2nm of Al to prevent oxidation, removed from the uhv system, transferred to another sample holder with electrical connections to allow the application of an electric field, and then reinserted into the uhv system. XAS, after reinsertion into the PEEM vacuum, confirmed that no oxidation of the Fe had occurred during the transfer of the sample from one cartridge to the other.

2. There is no explanation provided for why a thickness of 1.5 nm was chosen. On page 5, the authors mention that the behavior within a 1.5 nm Fe film differs from that of thicker films. However, they do not present any data on how the thicker Fe films behave. Furthermore, it is unclear how "thick" and "thin" are defined in this context—what is the boundary between them? It is also suggested that the authors investigate the effect of film thickness in relation to the proposed mechanism in the current work.

Our work concentrated on the ultrathin film regime on bulk BTO substrates, which has not been explored before. For Fe films as thin as 0.6 nm we observed large areas with no coupling to the underlying BTO FEL domains. For Fe films thicker than 4 nm, the Ti XAS signal level would have been much reduced so that imaging the underlying FEL domains would have been more difficult and so we did not explore this regime.

A full thickness dependent study would require a significant amount of additional beamtime and would only reveal results up to a thickness of ~4nm due the attenuation of the Ti signal with the increasing Fe thickness. Hence, and unfortunately, a thickness dependent study was outside the scope of our study, but we hope to inspire further x-ray microscopy studies now that we have established that ultrathin, metallic Fe films can be grown on BTO.

3. The experimental investigations appear to focus primarily on plane-view observations, with a lack of cross-sectional characterization of the samples. TEM could be employed to visualize stress or domain propagation, which would provide additional support for the authors' simulated results.

We thank the reviewer for the suggestion of using TEM to probe the interface. We have employed a wide range of techniques to understand the magnetoelastic properties of the metallic Fe film on BTO including PEEM, LEEM, LEED, XAS, XMCD, XRD, XRR, FMR along with micromagnetic simulations. We did not consider performing TEM, but we have added XRD results to section I of the SM to demonstrate strained, epitaxial growth of the Fe films. The demonstration that metallic films can be grown on BTO substrates will inspire further work on this system and the cross-sectional study on focus-ion beam etched slices would add further insights into the magnetoelastic coupling. We have also to consider that the preparation of TEM cross-sections can modify the strain state of the material, *i.e.* allows for relaxation, that is not present in our PEEM study. Unfortunately, TEM is now beyond the current study.

4. The manuscript would benefit from more effort in terms of organization, as the lack of subsection divisions makes it a bit harder to follow.

We thank the reviewer for this suggestion and we have added section headings to the main text.

5. The authors may consider measuring some macroscopic magnetoelectric coupling parameters to support their discussion, as they specifically mention the magnetoelectric coupling effect or coefficient in the introduction with regard to potential applications.

We thank the reviewer for their recommendation. We have added a new section to the SM (see SM section X) detailing a Ferromagnetic Resonance (FMR) study of an Al-capped sample showing results that are compatible with the conclusions of the PEEM study of an Al-capped sample. The value of B_2 is significantly reduced, with respect to the value for bulk *bcc*-Fe and positive, indicating the presence of the two FEL a_1 and a_2 domains

6. One might wonder how the electrode was deposited with a thickness of 2 nm, and whether this introduces any variables related to stress. Did the authors compare this electrode to others, such as Pt? This comparison could help exclude any additional variables introduced by the electrode, especially since it has the same thickness as the Fe film.

For soft X-ray XMCD a low Z metallic material is often preferred as the capping layer for a number of reasons. Firstly, the absorption coefficient is approximately proportional to Z^4 leading to two consequences: (i) high Z capping layers have higher absorption meaning fewer photons reach the buried magnetic material of interest and (ii) the increased absorption of X-rays by high Z capping layers leads to a higher background of photoemitted electrons. Small changes in the XAS at the absorption edge of interest, especially in dilute systems, are harder to measure on signals with high backgrounds. Secondly, materials such as Pt, which have large

spin-orbit coupling strengths, *may* hybridize with the magnetic material (especially if ultrathin) modifying the magnetic properties. We have added a sentence to the Methods section to explain the use of Al. We agree that the Al electrode does play a role in affecting the ME properties of the film. The samples that were capped with Al, to allow for the application of an electric field, yielded a slightly positive magnetoelastic constant B_2 , but still much lower than the value for bulk *bcc*-Fe.

Heavier elements (e.g. Pt) also have a high spin-orbit coupling, which could lead to hybridization with the Fe film, altering its magnetic properties. Al is also particularly effective at protecting the underlying Fe film because the oxidation of the topmost atomic Al layers prevents diffusion of atmospheric oxygen into the bulk (unlike Cu which oxidizes all the way through the film over time).

7. I believe the authors mention the polycrystalline nature of the Fe films. However, in the conclusion paragraph (page 10), they summarize the manuscript by referring to epitaxial growth, which is quite confusing. This conclusion is difficult to follow, especially in the context of the key highlights of the current work. Additionally, how should the epitaxial growth of metals on perovskite inorganic materials be defined? Are the Fe films polycrystalline, epitaxial, or amorphous? There is a lack of convincing discussion on this point.

The Fe films in our study grew epitaxially, as confirmed by the additional XRD and LEED data now presented in section I of the SM. Epitaxial growth of Fe on other inorganic oxide surfaces, such as MgO, is common and related to the relative crystal symmetries and lattice constants – see Fig.5 in APL 118, 042411 (2021) for example. In our case the lattice mismatch between Fe and BTO is around 1.6% ($\text{Fe } 2.87\sqrt{2} \text{ \AA} \approx \text{BTO } 3.99 \text{ \AA}$) which favours epitaxial growth of Fe on BTO, see Fig. SM 1. There is a reference to a polycrystalline film in the introduction, but this is in relation to a previous study (Phys. Rev. Materials **3**, 024403 (2019)) which is unrelated to our study with respect to the samples used. We have amended the sentence to try to make the distinction between the two studies clearer. We have also added more information regarding the sample growth to the Methods section.